# Mechanical Strength and Hydration Characteristics of Cement Mixture with Highly Concentrated Hydrogen Nanobubble Water

**DOI:** 10.3390/ma14112735

**Published:** 2021-05-22

**Authors:** Won-Kyung Kim, Gigwon Hong, Young-Ho Kim, Jong-Min Kim, Jin Kim, Jung-Geun Han, Jong-Young Lee

**Affiliations:** 1School of Civil and Environmental Engineering, Urban Design and Study, Chung-Ang University, Seoul 06974, Korea; kwonk2004@naver.com (W.-K.K.); younghogeo@naver.com (Y.-H.K.); 2Department of Civil and Disaster Prevention Engineering, Halla University, Wonju-si 26404, Korea; g.hong@halla.ac.kr; 3School of Mechanical Engineering, Chung-Ang University, Seoul 06974, Korea; 0326kjm@cau.ac.kr; 4Department of Intelligent Energy and Industry, Chung-Ang University, Seoul 06974, Korea; kimjin4311@nate.com

**Keywords:** hydrogen nanobubble water, mortar, mechanical strength, hydration

## Abstract

In this study, highly concentrated hydrogen nanobubble water was utilized as the blending water for cement mortar to improve its compressive and flexural strengths. Highly concentrated nanobubbles can be obtained through osmosis. This concentration was maintained by sustaining the osmotic time. The mortar specimens were cured for 28 days, in which the nanobubble concentration was increased. This improved their flexural strength by 2.25–13.48% and compressive strength by 6.41–11.22%, as compared to those afforded by plain water. The nanobubbles were densified at high concentrations, which caused a decrease in their diameter. This increased the probability of collisions with the cement particles and accelerated the hydration and pozzolanic reactions, which facilitated an increase in the strength of cement. Thermogravimetric analysis and scanning electron microscopy were used to confirm the development of calcium silicate hydrate (C-S-H) and hydration products with an increase in the nanobubble concentration. Quantitative analysis of the hydration products and the degree of hydration were calculated by mineralogical analysis.

## 1. Introduction

Concrete is one of the most extensively used building materials. As it is almost impossible to replace concrete with other materials from natural resources, the use of cement is inevitable [1,2]. However, the energy consumption of the cement industry is high and carbon dioxide (CO_2_) emissions from Portland cement (a commonly used type of cement) production cause adverse environmental problems. For every ton of cement produced, approximately 900 kg of CO_2_ is discharged into the atmosphere. Furthermore, approximately 2/3 of the CO_2_ emissions in the world originate from cement factories [3,4]. Both reducing the consumption of cement by substituting cement with slag or pozzolan and improving the function of blended water have been considered as solutions for sustainable development [5,6]. Recently, numerous studies have focused on improving the properties of concrete by applying nanotechnology to architectural structures [7,8]. Kim et al. improved the compressive strength of cement mortar specimens using functionalized graphene oxide and hydrogen nanobubble water (HNBW) [9]. Zadeh et al. studied the mechanical strength and durability of concrete applied with zeolite (aggregate), meta-kaolin (pozzolan), and micro-nano bubble water (mixed water) [10]. Han et al. improved the mechanical properties and durability of cement mortar by using HNBW. In this study, the concentration of HNB was reduced by using the natural disappearance of bubbles, and the bubble’s diameter was about 200–400 nm. As the concentration of HNB was halved, it was experimentally confirmed that the increment in mortar compressive strength decreased from about 17% to about 4% in 7 days of curing [11].

Bubbles are generated via the dispersion of gas in the liquid [12]. In general, coarse bubbles with a size of 50 µm or more are called macrobubbles, bubbles of sizes between 10 and 50 µm are called microbubbles, and 200 nm-sized bubbles are called nanobubbles [13]. Nirmalkar et al. experimentally proved that nanobubbles generated by ultrasonic cavitation at 20 kHz exist in a bulk form filled with gas and survive in water for about 10 months [14]. A microscopic size, high internal pressure, and negative surface charge enhance the stability of nanobubbles in water [15]. Owing to their small size, nanobubbles with a low emerging speed and high zeta potential values do not undergo coalescence or aggregation, due to inter-bubble repulsion, and the bubble behavior is affected by Brownian motion and does not disappear even after emerging on the surface [16,17,18,19]. The lifespan of a nanobubble depends on the rate at which it is formed and the dissolution and diffusion of the gas into the bulk solution [20,21,22,23]. Gas dissolution and diffusion are affected by several factors such as gas solubility, temperature, bubble size, pressure difference, and gas saturation of the solution [24]. The smaller the size, the more proficient the increase in internal pressure. Therefore, the gas dissolves or diffuses into the bulk solution at a higher rate. Ohgaki et al. produced a nanobubble with an average radius of 50 nm, and the internal pressure was measured as high as about 6 MPa [22]. Weijs et al. numerically confirmed that the bulk nanobubbles dissolved in water can be stable for longer when it is dispersed in a cluster form [23]. At this moment or stage, strong hydrogen bonding on the bubble surface resists the increasing internal pressure and prevents the bubble from collapsing. Therefore, nanobubbles can exist at significantly high concentrations in the solution [21] in the form of a cluster, reducing the rate of diffusion and dissipation of the gas and enhancing its stability [22]. Due to the specific properties of these nanobubbles, they have numerous applications [25,26] such as surface coating and cleaning [27], contaminant removal [28], energy system improvement [29], medicine [30], fluid engineering [31], agriculture, and vegetable growth [32]. Furthermore, the stability of a hydrogen nanobubble (HNB) is maintained because the interface curvature is lower than expected due to the high contact angle [24]. During the mineral flotation of fine and ultrafine particles, nanobubbles have been shown to increase the contact angle (coal, phosphate, quartz, and copper minerals) to improve particle–bubble adhesion [33].

In this study, highly densified HNBW, which contained uniformly sized bubbles (<200 nm) with high stability in water, was gradually obtained via osmosis. Then, HNBW with various levels of concentrations was applied as a mixing solvent for the production of cement mortar to improve its mechanical strength and durability. Subsequently, the change in the hydration characteristics of HNBW was confirmed through X-ray diffraction analysis (XRD) and thermogravimetric analysis (TGA) of the cement paste formed using different concentrations of HNBW, while maintaining the same mixing ratio. Moreover, scanning electron microscopy (SEM) images were captured to analyze the effect of the concentration of nanobubbles on the development of hydration products and to assess the impact of the change in the hydration properties of hardened cement mixtures on their mechanical strength.

## 2. Experimental Details

### 2.1. Synthesizing the HNBW

In this study, to produce HNBW containing bubbles with long-term stability, a capillary via a decompression method was applied [34]. By measuring the concentration and size of the nanobubbles for approximately 30 days, it was confirmed that a concentration of approximately 100 million bubbles per unit volume was presented and maintained. The size distribution and concentration of the nanobubbles were measured through nanoparticle tracking analysis (NTA, LM10, Malvern Panalytical, Westborough, MA, USA). Furthermore, osmosis was applied to obtain highly concentrated HNBW [35]. Figure 1 shows the principle of osmosis applied by the high concentration method. Approximately 60 wt.% of polyethylene glycol (PEG, MW = 20,000, CP, DAEJUNG, Korea) aqueous solution was used as the hypertonic solution, and HNBW, used as a hypotonic solution, was placed in a semi-permeable pocket to sustain the osmosis process for intervals of approximately 40 min [36]. Figure 2 shows the change in the volume of the semi-permeable pocket corresponding to osmosis duration and the decrease in the volume as the osmosis time progressed. This decrease in the volume occurs because water molecules inside the pocket pass through the pores of the semi-permeable membrane. Table 1 presents the results of the NTA employed to measure the size and concentration of the bubbles. Figure 3 shows the bubble size distribution and the image of concentrated HNBs measured by NTA. As the osmosis duration increased, the concentration of the HNBs gradually increased to approximately 1.5 to 3 times higher than the initial bubble concentration. However, the diameters of the HNBs gradually decreased from approximately 200 to 150 nm and were subsequently equalized.

### 2.2. Experimental Method and Conditions

Table 2 shows the mix proportions for each test conducted in the present study based on the HNBW concentration and the conditions used for preparing the specimens for the XRD, TGA, and SEM analyses. The materials used for the preparation of the specimens were Ordinary Portland cement (OPC) and ISO standard sand. In ISO 679, sand with a wide particle size range of 0.08–2.0 mm is specified as standard sand [37]. Plain water was used as the compounding solvent in Case A, whereas low- and high-concentration HNBW were used for Cases B and C, respectively.

The cement mortar was subjected to mechanical strength evaluations and SEM analyses. All the specimens (i.e., those in Cases A, B, and C) were prepared according to ISO 679 standards and were cured for 3, 7, and 28 days, respectively, in clean water at 20 °C [37]. The mechanical strength of the mortar was measured using a flexural and compressive test device for mortar beams (HJ-1295, Heungjin Testing Machine, Gimpo, Korea). The deformation rate was set to 1 mm/m, and three specimen samples were selected for each experimental condition. After they were cracked, the average flexural strength and compressive strength were evaluated.

Cement paste specimens for analyzing hydration characteristics were prepared according to ASTM C109/C109M and were cured with water for seven days [38]. XRD (MiniFlex 600, Rigaku Co., Tokyo, Japan) analysis was performed at diffraction angle intervals of 0.02° over a range of 5–90°. Specimens in the form of a lump of approximately 30×30 mm were used. TGA (Q500, TA Instrument, Inc., New Castle, DE, USA) was carried out by heating the specimens to approximately 1000 °C at a heating rate of 5 °C/min for about 50 g of sample per condition. In addition, the experiment was conducted under an inert atmosphere (nitrogen, N_2_(g)), which shows low reactivity with other substances. All samples used in the XRD and TGA analysis were cured for seven days and were impregnated with acetone for at least five days to prevent curing and carbonation.

## 3. Results and Discussion

### 3.1. Mechanical Strength Test

Figure 4 shows the mechanical strengths of the cement mortar specimens. The average flexural and compressive strengths of the specimens in Cases B and C, which involved the use of stabilized HNBW as the blending solvent, were superior to those of the specimen in Case A. Moreover, as the concentration of HNBs increased, the mechanical strength increased. Table 3 and Table 4 present the average values of the flexural strength and compressive strength for each curing duration and condition. In the case of curing for 28 days, compared to that of Case A, the flexural strength in Case B showed a very slight improvement of 2.25%. On the other hand, in Case C, it showed a higher improvement of 13.48%. Similarly, the compressive strengths in Case B and C exhibited improvements of 6.41% and 11.22%, respectively, as compared to that of Case A.

As discussed in Section 2.1, it was proved that the diameter of the HNBs decreased with the concentration. This reduction in diameter contributes toward the stability of the HNBs, which affects the activation of the cement hydration reaction [10]. Reducing the bubble diameter can increase the probability of collisions between the bubble and cement particle. Unhydrated cement particles are known to possess a lower floating ability, as they are considerably small in size (20–30 µm). However, flotation can be improved by increasing the probability of collision between the bubble and the cement particle [10,39]. The collision probability is expressed in Equation (1).
(1)$$P_{c}=\left[3/2+\left(4R_{eb}{}^{0.72}\right)/15\right]{\left(D_{p}/D_{b}\right)}^{2}$$
where *D_p_*, *D_b_*, and *R_eb_* represent the particle diameter, bubble diameter, and Reynolds number, respectively. Figure 5 shows the simulation results of Equation (1) assuming the bubble diameter *D_b_* = 100–700 nm, which are the results of the HNB size distribution obtained through NTA. In addition, the particle diameter *D_p_* was set to 10–40 µm by applying the size of the unhydrated cement particles dispersed in water. Another parameter, the HNBW’s Reynolds number *R_eb_*, was assumed to be 100. The collision probability *P_c_* increased exponentially as *D_b_* decreased in the size reduction range (100–200 nm) of HNB concentrated through osmosis. As a result, it is observed that collision probability of the fine particles such as cement particles can be improved by shrinking the bubble diameter [40,41].

As illustrated in Figure 6, very fine particles with nanobubble-coated surfaces can lead to particle aggregation. They can easily adhere to entrapped air bubbles (>1 mm) that are conventionally formed during mixing of the cement mixture [42], and this phenomenon increases the overall flotation [33]. Increasing the collision between bubble and cement particle increases the degree of hydration reaction of the particles; this helps form a homogeneous mixture, wherein the hydrates are tightly packed [10]. In other words, it can be said that the HNBs, whose diameters were reduced via osmosis (i.e., their concentration increased), effectively activated the hydration reaction by increasing the number of collisions with the cement particles, thereby resulting in an improvement in the cement mortar strength.

### 3.2. Mineralogical and Thermal Analyses

#### 3.2.1. XRD Test

Figure 7 shows the XRD patterns for the different mixing conditions of the cement paste cured in water for 7 days. In all cases, major hydration products such as portlandite (Ca(OH)_2_, CH), ettringite, unhydrated clinker minerals, and calcite (CaCO_3_) were observed; the peak corresponding to CH was particularly strong. CH and ettringite were produced via an active hydration reaction at the initial stages of curing [43]. During underwater curing, CH collectively dissolved in water, and calcium ions (Ca^2+^) were formed. It was determined that a small amount of carbon dioxide (CO_2_) was released into the atmosphere and reacted with calcium ions to generate calcite [44]. In the case of calcium silicate hydrate (C-S-H), which plays a crucial role in enhancing the strength of the cement mixture, the crystal phase was not observed in the XRD pattern, because it was amorphous and irregular [43,45]. 

By comparing the XRD patterns of the cement paste, Cases A, B, and C shared similar patterns. As the HNB concentration increased, a slight difference in peak intensity was observed even though there was no difference in the type of the crystal phase. Consequently, it was confirmed that the new crystal phase was not formed by the HNBs. In addition, it was confirmed that the reaction rate of HNBs with cement particles influenced the improvement of mortar strength.

#### 3.2.2. TGA Test

##### Methodology

TGA is the most widely used method to determine the degree of hydration (α) [46]. This approach identifies cement hydrates by measuring the weight loss occurring within a specific temperature range. This is required to determine the amount of water (*W_B_*) that is chemically bound to the hydrate in the cement mixture, which is used to estimate the degree of hydration [47]. In this study, the degree of hydration was calculated by applying Bhatty’s method, according to the formula given below [48]:(2)$$W_{B}=Ldh+Ldx+0.41\left(Ldc\right)$$
(3)$$\alpha =W_{B}/0.24$$
where *Ldh*, *Ldx*, and *Ldc* refer to the relative mass losses indicated by the TGA curves due to the dehydration of C-S-H, dehydroxylation of CH, and decarbonization of CaCO_3_, respectively. The decomposition of cement hydrates by heat involves three main steps: weight loss in the temperature range of 25–400 °C, indicating the evaporation of free water (25–105 °C) and dehydration (*Ldh*) of C-S-H (105–400 °C); dehydroxylation (*Ldx*) of portlandite occurring in the temperature range of 400–600 °C; and decarbonization (*Ldc*) of CaCO_3_ occurring in the temperature range of 600–800 °C. The constant value of 0.41 in Equation (2) is the coefficient of change for calculating *W_B_* derived from carbonated CH. In addition, the constant value 0.24 in Equation (3) represents the maximum *W_B_* required to hydrate the entire cement particle, which can vary from 0.23 to 0.25. However, in this case, 0.24 was used for the OPC. Table 5 shows the temperature ranges and types of hydrates according to Bhatty’s method [48].

##### Test Results

Each sample, with a mass of 50 mg, was heated as follows: (i) maintained at 28 °C for 10 min; (ii) heated up to 105 °C with a heating rate of 5 °C/min; (iii) maintained for 30 min; and (iv) heated up to 1000 °C with a heating rate of 5 °C/min [49]. The inner space of the chamber was filled with pure nitrogen (N_2_) gas flowing at 20 mL/min. The temperature was maintained for a certain period during heating to remove the free water inside the hardened cement paste. TGA test results obtained using Bhatty’s method are shown in Figure 8. 

During initial heating, that is, from 28 °C to 105 °C, the weight reduction in all cases was similar. However, as the concentration of bubbles increased while heating up to 1000 °C, the weight reduction gradually increased; this is reflected by the increase in the difference between the curves. Furthermore, a sudden change in curvature was observed for each temperature range, as mentioned in Table 5. Table 6 shows the residual weight of the cement paste for the corresponding temperature, as determined via TGA. Table 7 presents the percentages of the chemically bonded water (*W_B_*) and the degree of hydration (α) based on the weight reduction at different temperature ranges.

##### Discussion

The increment in relative mass loss (Δ*Ldh*, Δ*Ldx*, and Δ*Ldc*) of each cement hydrate compared to Case A is shown in Figure 9. This shows that the amount of cement hydrate increased with a higher concentration of HNB. Overall, the increment in CaCO_3_ (Δ*Ldc*) was similar in Cases B and C, at 2.24% and 2.32%, respectively. On the other hand, the increases in C-S-H (Δ*Ldh*) and CH (Δ*Ldx*) affected the difference in the total amount of cement hydrate produced in Cases B and C. In particular, the increase in Δ*Ldh* was greater than that of Δ*Ldx*. This means that C-S-H (3CaO∙SiO_2_∙3H_2_O) developed more actively than CH (Ca(OH)_2_) according to bubble concentration.
(4)$$2\left(3\mathrm{CaO}\cdot {\mathrm{SiO}}_{2}\right)+6H_{2}O\to 3\mathrm{CaO}\cdot 2{\mathrm{SiO}}_{2}\cdot 3H_{2}O+3\mathrm{Ca}{\left(\mathrm{OH}\right)}_{2}$$
(5)$$2\left(2\mathrm{CaO}\cdot {\mathrm{SiO}}_{2}\right)+4H_{2}O\to 3\mathrm{CaO}\cdot 2{\mathrm{SiO}}_{2}\cdot 3H_{2}O+\mathrm{Ca}{\left(\mathrm{OH}\right)}_{2}$$

Equations (4) and (5) represent the hydration reaction of C_3_S (3CaO∙SiO_2_) and C_2_S (2CaO∙SiO_2_), the main components of cement, respectively. As mentioned in Section 3.1, the improvement of bubble–particle collision probability and flotation due to the high concentration and shrinkage of bubbles increased the contact area with water, resulting in the development of C-S-H and CH compared to Case A.
(6)$$3\mathrm{Ca}{\left(\mathrm{OH}\right)}_{2}+2{\mathrm{SiO}}_{2}\to 3\mathrm{CaO}\cdot 2{\mathrm{SiO}}_{2}\cdot 3H_{2}$$
(7)$$\mathrm{Ca}{\left(\mathrm{OH}\right)}_{2}+{\mathrm{CO}}_{2}\to {\mathrm{CaCO}}_{3}+H_{2}O$$

The CH generated by active hydration is consumed in the pozzolanic reaction to form C-S-H and the carbonation to form CaCO_3_, and the respective chemical equations are shown in Equations (6) and (7). Due to the HNBs, blended water is widely dispersed inside the cement mixture, resulting in a homogeneous mixture. During this process, as additional cement particles come into contact with water molecules, reactive components such as calcium ions (Ca^2+^) are eluted and the overall hydration improves, thereby enhancing the CH content [50]. Even if CH is consumed during pozzolanic and carbonation reactions, CH is continuously produced through a hydration reaction promoted by HNB. This improves the CH content slightly, but not as much as the C-S-H content. Furthermore, the results of this experiment are similar to those reported by Feng et al. [43], who used TGA to evaluate Portland limestone cement.

Therefore, the increment in C-S-H was greatest due to the activation of hydration and pozzolanic reactions. Compared to Case A, the C-S-H content increased by approximately 1.41% in Case B and 2.78% in Case C. When the population of HNB is increased, the C-S-H content also increased, causing a gradual improvement in mechanical strength, as explained in Section 3.1. In the hardened cement paste, C-S-H occupies a large surface area; it acts as the hydration product that densely fills the internal spaces to improve the durability of the mixture [49]. Kim et al. experimentally confirmed the pore size distribution and porosity in the hydrate due to the high concentration of HNB through mercury intrusion porosimetry (MIP). In this study, as the CH (about 1 µm) was gradually consumed and became an irregular shape, C-S-H (<0.1 µm) gradually developed, resulting in a smaller internal pore size and improved watertightness and durability [51]. The hydration degree (α) of the cement paste cured in water for seven days, as determined via Bhatty’s method, was 46.56% for Case A, 57.97 % for Case B, and 67.17 % for Case C. Compared to Case A, the hydration degrees for Cases B and C showed improvements of 11.41% and 20.61%, respectively.

In conclusion, it was observed that as the concentration of bubbles increased, the degree of weight reduction due to heat gradually increased. Furthermore, it was confirmed that the hydration and pozzolanic reactions were continuously promoted by the HNBW, leading to an increase in the overall α, which, in turn, improved the mechanical strength of the cement mortar. Moreover, the presence of HNBs improved the reactivity between cement particles, leading to the development of hydrates such as C-S-H. Consequently, the structure of the hardened mixture gradually became more tightly packed.

### 3.3. Microstructural Analyses

Figure 10 shows the SEM (S-3400N, Hitachi, Tokyo, Japan) images of the cement mortar prepared using plain water (Case A), stabilized HNBW (Case B), and high-concentration HNBW (Case C). The internal microstructure of each specimen was analyzed after 3, 7, and 28 days of curing. In Case A, various hydration products such as portlandite (CH), ettringite, and C-S-H, which were formed via active hydration reactions at the initial stages of curing, were observed. However, the formation of the hydration products was slower than those in Cases B and C, and a larger number of internal pores were observed. Ettringite was observed in all cases, and it was confirmed that the inner pores were densely filled with hydrates as the duration of curing increased. Furthermore, an increase in the concentration of HNBW enhanced the formation of ettringite and C-S-H crystals, even though the curing period remained the same. In Case B, many irregularly shaped CH crystals were distributed; the shape of CH became irregular due to subsequent conversion such as carbonation and pozzolanic reaction. As CH in the form of hexagonal crystals produced by the hydration reaction was consumed from its surface, the irregularly shaped CH crystallites oriented statistically randomly. In conclusion, this was attributed to the reactivity of CH with cement particles and the activation of the pozzolanic reaction by HNBs. Due to the activated reactions, in Case C, it was observed that internal pores were formed tightly with hydrates such as C-S-H. The acceleration of cement hydration by HNBW resulted in the development of hydrates, and the structure of the hardened body gradually became denser. Analyses of the XRD, TGA, and SEM images confirmed the improvement in mechanical strength and the hydration characteristics of the hardened structure of the cement mixture with highly concentrated HNBW.

## 4. Conclusions

In this study, the effect of the concentration of HNBs on the mechanical strength of cement mortar was tested. When the concentration of these HNBs was enhanced via osmosis and they were used as the blending solvent for preparing cement mortar, not only did the flexural strength increase by 13.48% in 28 curing days, but its compressive strength also increased by 11.22% in 28 curing days, as compared to the strengths based on plain water as the blending solvent. This improvement is attributed to the high concentration of nanobubbles with small diameters. These nanobubbles enhance the collision between the solvent and cement particles, accelerating the hydration and pozzolanic reactions.

Furthermore, TGA was used to confirm the increment in the amount of CH in the specimen with HNBW. This increment enhanced the contact between cement and water molecules because of the high concentration of nanobubbles. Reactive components such as calcium ions (Ca^2+^) were eluted when the hydration reaction was activated. When HNBW was used, the C-S-H ranged from 1.41% to 2.78%, while the hydration degree was 11.41–20.61%, as compared to those for the specimen with plain water. The development of C-S-H and hydration products led to the formation of a dense structure, which was confirmed through SEM analyses.

The high concentration and size control technique of nanobubbles with long-term stability in water used in this study will be applicable as a core technology in various industrial sites such as wastewater purification, aquaculture, and construction materials. Particularly, research on sustainable building materials with nanotechnology using eco-friendly materials such as micro-nanobubble water is steadily active. However, additional research is needed to apply it to the construction site, such as a mass production technique of nanobubble water and a high concentration method that can be realized in the field.

## Figures and Tables

**Figure 1 materials-14-02735-f001:**
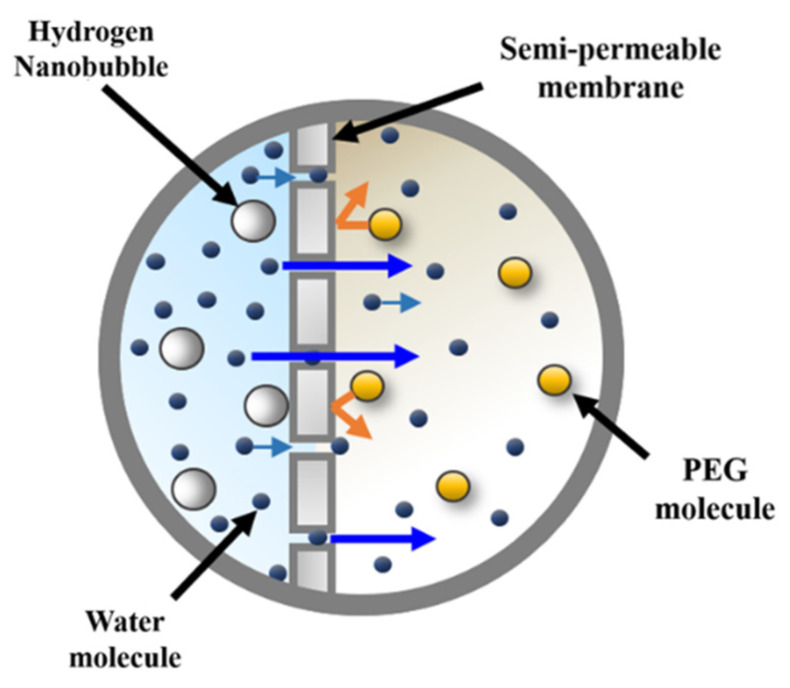
Schematic view of high concentration of nanobubbles using osmosis.

**Figure 2 materials-14-02735-f002:**
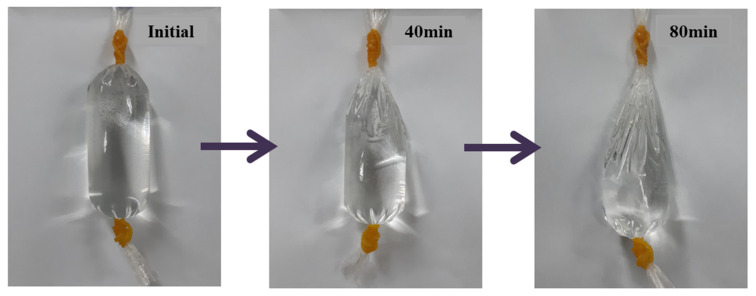
Decrease in volume of the cellophane pocket as osmosis continued.

**Figure 3 materials-14-02735-f003:**
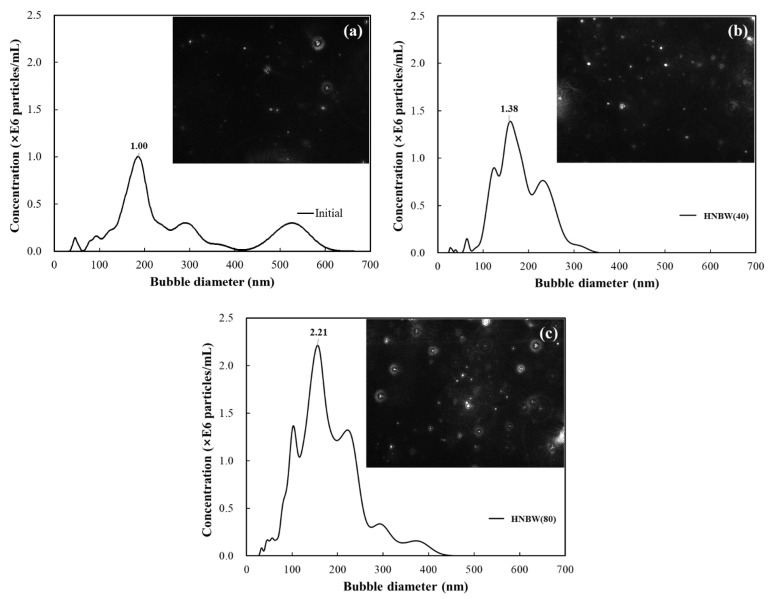
NTA test results for size distribution and image of nanobubbles: (**a**) Initial; (**b**) HNBW (40); (**c**) HNBW (80).

**Figure 4 materials-14-02735-f004:**
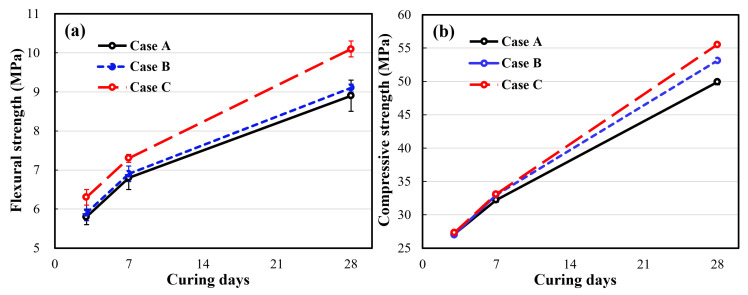
Mechanical strength of cement mortar (ISO 679): (**a**) average flexural strength; (**b**) average compressive strength.

**Figure 5 materials-14-02735-f005:**
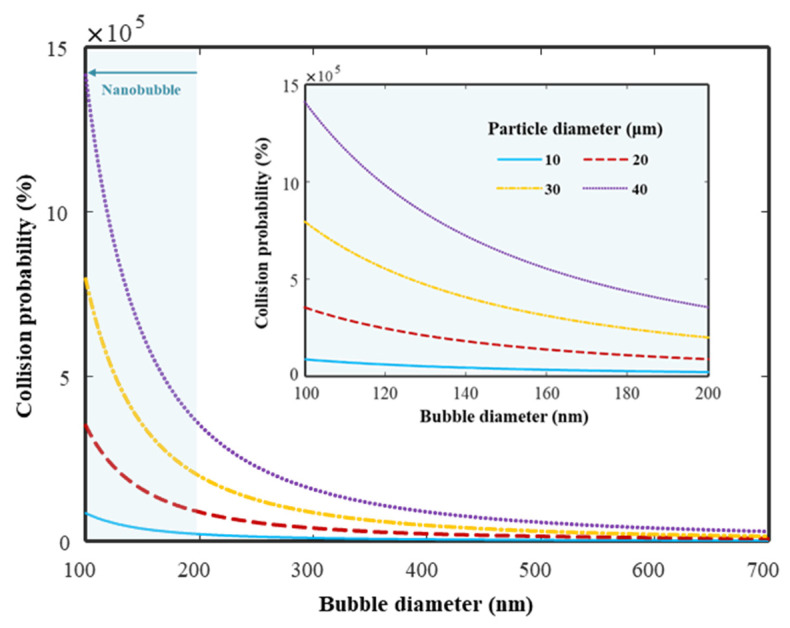
Bubble–particle collision probability by applying typical cement particle size and HNB size measured by NTA: bubble diameter *D_b_* =100–700 nm, particle diameter *D_p_* =10–40 µm, bubble’s Reynolds number *R_eb_* =100. Inset shows a graph for the HNB diameter section (100–200 nm) reduced with high concentration.

**Figure 6 materials-14-02735-f006:**
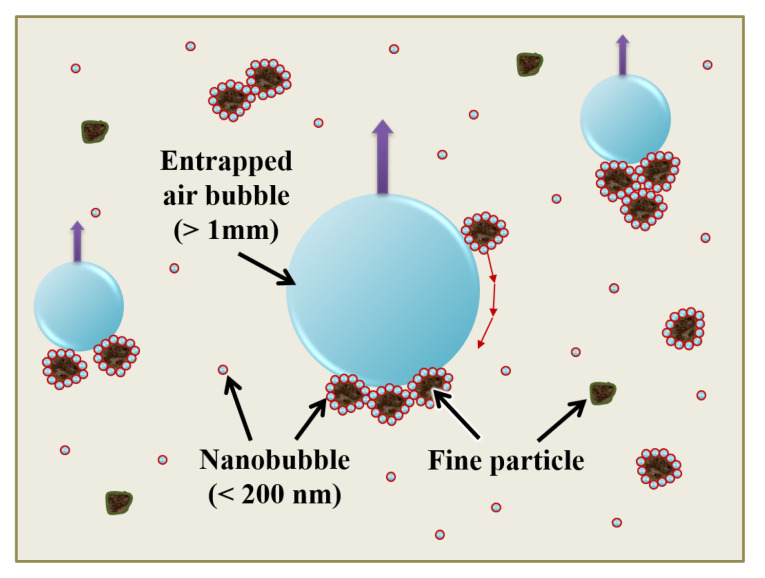
Increased bubble–particle collision probability and flotation of very fine particles.

**Figure 7 materials-14-02735-f007:**
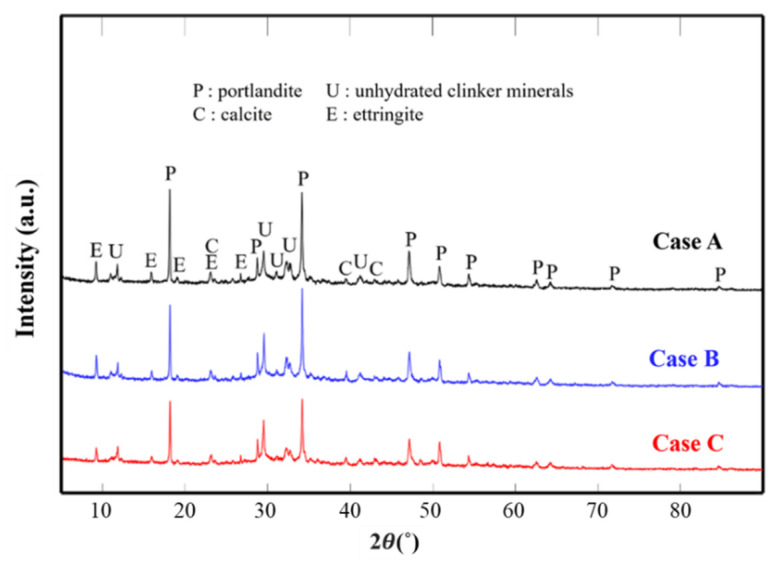
XRD patterns for the three cement pastes prepared using different concentrations of HNBW.

**Figure 8 materials-14-02735-f008:**
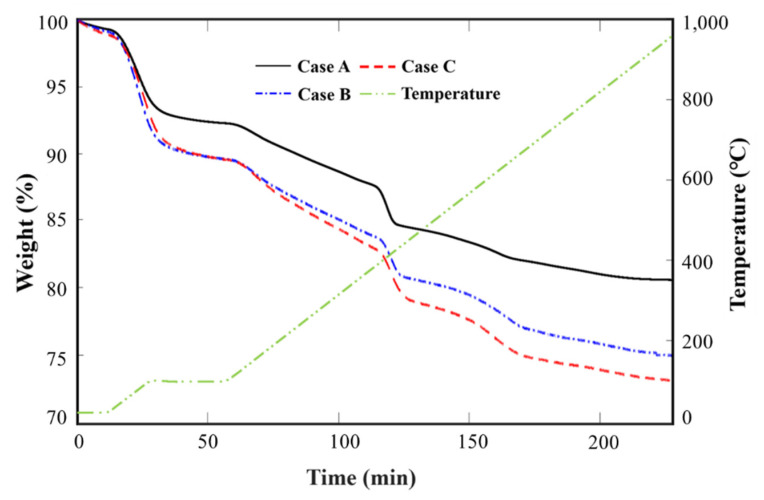
TGA curves of the three types of cement paste based on different concentrations of HNBW.

**Figure 9 materials-14-02735-f009:**
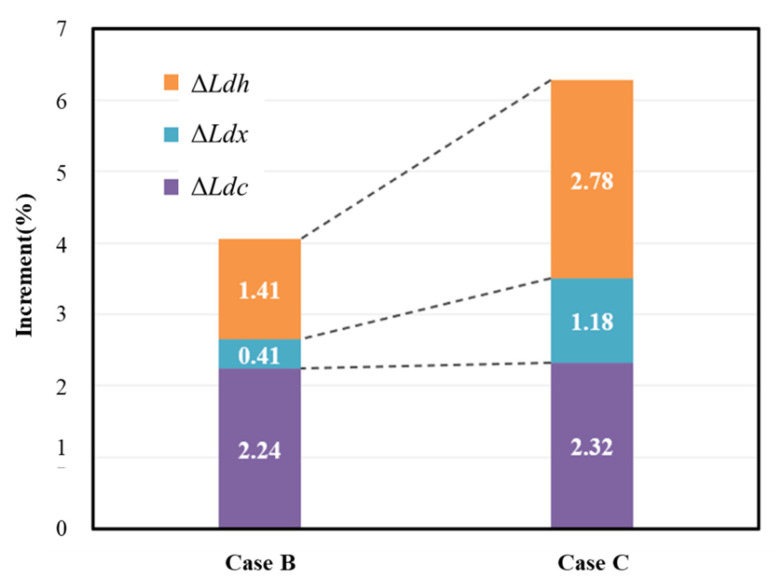
Increments in *Ldh*, *Ldx*, and *Ldc* compared to Case A.

**Figure 10 materials-14-02735-f010:**
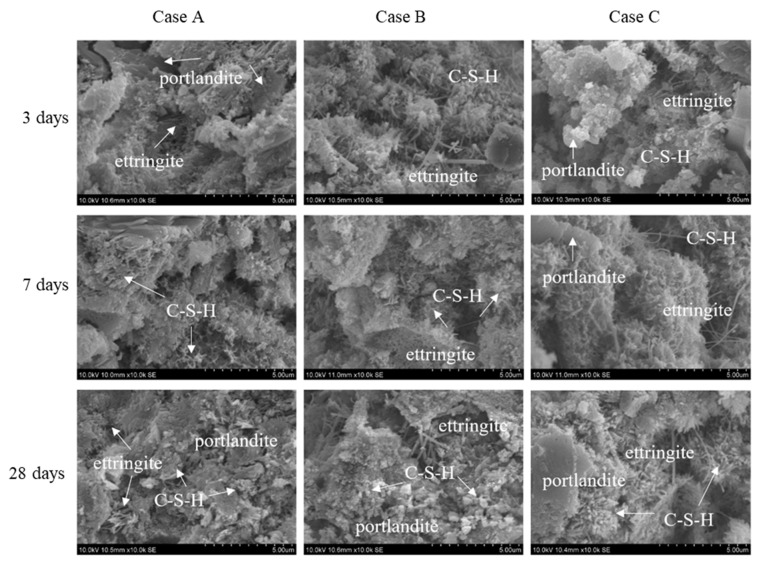
SEM images for cement mortar based on three types of blending solvents for different periods of curing (3, 7, and 28 days).

**Table 1 materials-14-02735-t001:** NTA results according to HNB concentrations.

Properties	Initial	HNBW (40)	HNBW (80)
Osmosis duration (min)		40	80
Concentration (particles/mL)	HNBW (0.93 × 10^8^)	HNBW (1.36 × 10^8^)	HNBW (2.41 × 10^8^)
Mean diameter (nm)	199.33	177.33	153.01
Mode diameter (nm)	179.67	146.67	135.25

**Table 2 materials-14-02735-t002:** Experimental conditions.

Mixtures	Mixing Ratio (%)	Sample No.			Curing Time (day)	Test Method *
		Case A	Case B	Case C		
Cement mortar (ISO 679)	22.2	Ordinary Portland cement (OPC)			3, 7, 28	①, ②, ③
	66.7	ISO standard sand				
	11.1	Plain water	HNBW (40)	HNBW (80)		
Cement paste (ASTM C 109)	67.2	Ordinary Portland cement (OPC)			7	④, ⑤
	32.8	Plain water	HNBW (40)	HNBW (80)		

***** ① Flexural strength test, ② Compression strength test, ③ SEM (scanning electron microscope), ④ XRD (X–ray diffraction test), ⑤ TGA (thermogravimetric analysis).

**Table 3 materials-14-02735-t003:** Average flexural strength (unit: MPa).

	Curing Days	3	7	28
Sample No.				
Case A		5.8	6.8	8.9
Case B		5.9	6.9	9.1
Case C		6.3	7.3	10.1

**Table 4 materials-14-02735-t004:** Average compressive strength (unit: MPa).

	Curing Days	3	7	28
Sample No.				
Case A		27.1	32.2	49.9
Case B		27.0	33.0	53.1
Case C		27.3	33.1	55.5

**Table 5 materials-14-02735-t005:** Temperature ranges of Bhatty’s method [48].

Temperature Range (°C)	Region	Decomposition of Cement Hydrates
105–440	Dehydration (*Ldh*)	C-S-H
440–580	Dehydroxylation (*Ldx*)	Ca(OH)_2_
580–1000	Decarbonization (*Ldc*)	CaCO_3_

**Table 6 materials-14-02735-t006:** Mass loss during TGA tests.

Mixture	M_105 ˚C_ (mg)	M_440 ˚C_ (mg)	M_580 ˚C_ (mg)	M_1000 ˚C_ (mg)
Case A	9.93	9.11	8.96	8.67
Case B	9.79	8.84	8.66	8.19
Case C	9.78	8.70	8.46	7.99

**Table 7 materials-14-02735-t007:** Ldh, Ldx, Ldc, W_B_, and α calculated using TGA (Bhatty’s method).

Mixture	*Ldh* (%)	*Ldx* (%)	*Ldc* (%)	*W_B_* (%)	α (%)
Case A	8.27	1.58	3.23	11.17	46.56
Case B	9.68	1.99	5.47	13.91	57.97
Case C	11.08	2.76	5.55	16.12	67.17

## Data Availability

Not applicable.

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
