# Peer review of "Mechanical Strength and Hydration Characteristics of Cement Mixture with Highly Concentrated Hydrogen Nanobubble Water"

_materials, 2021, doi:10.3390/ma14112735_

Round 1
Reviewer 1 Report
Dear Authors,
your idea for this paper is simple, but efficiently and clearly presented. Your conclusions are supported by your research.
Best regards, Reviewer
Should be correct:
pages should be "full" (see p. 5, 7 11)
figures should be wider or centered
table 3 and 4 one cell is crossed - why?
Reviewer 2 Report
This study deals with the effect of the concentration of hydrogen nanobubble water on the mechanical strength of cement mortar. The results were supported by TGA, XRD, and SEM. The quality of experimental results as well as attention to details such as ASTM standards, sample preparation, etc are impressive. However, there are some issues in the text that needs to be addressed before publication.
- Line 43-44, please rephrase “metakaolin (cement)”. Metakaolin is characterized as a pozzolan, not cement.
- Line 47, 52, 62; please exemplify the level of influence, time, and the rate for clarity of the statements.
- Line 122, please provide size characteristics of the sand used.
- Line 185 and Figure 6; the term “conventional sized air bubble” and “conventional air bubble” are misleading. These bubbles are formed by the conventional mixing technique. Please rephrase them.
- Section 3.1. Although the sample preparation was according to ASTM standard and the trend of changing strength is clear, the limited number of data points and also reporting only the average value put doubts on the validity of the results. Please draw all three points for each sample on the curve or provide max/min in the table to clarify the results.
- Section 3.2. Please provide information about which samples (3 or 7 or 27 days) were used for TGA and XRD.
- Section 3.2.1. Many instrumental factors affect the height of the peaks in XRD. Without a reference peak such as that of the Carbon or SiO2 sample, height comparison is not reliable among different samples. As the content of each phase can also change the height, the discussion about the height of the peaks should be backed by their intensity compared to a reference peak in the first place and then, by other discussions or characterization techniques.
- Section 3.2.2. Using Ldh, Ldx, and Ldc in the text makes the whole discussion (3) confusing. CSH and CH components are very known materials and utilizing their acronym in the text would clear the text from confusion.
In addition, there are some grammatical errors in the text. Please address the following comments;
- Line 77, subsequently instead of consequently
- Line 35, 69, 75, 138, and 243; “further” is not a suitable choice for these sentences. Please replace them with furthermore, then, etc. depending on the sentence
- Line 81, passive instead of active voice for “capture”
- Line 133, a verb like “prepared” is missing.
Reviewer 3 Report
Manuscript deals with interesting topic dealing with properties of cement mixture with highly concentrated hydrogen nanobubble water. However, the manuscript brings new findings to this topic, which can certainly be described as interesting. But I see two problems in the presented study:
1) In practice (use on site), the creation of HNBW containing bubbles with long-term stability will be very difficult, especially in terms of price for the large amount needed for the building structure or other uses in industry. This is the biggest weakness of the whole technology. Is it even possible to use it outside the laboratory? I lack a detailed analysis of this problem in the manuscript. I urge this to be added and described extensively. Otherwise, it is just “research for research”.
2) The result in chapter 4. Conclusion that the compressive strength of the tested cement mortar was increased by “2.25%-14.48%” is slightly controversial. The measured increase in strength by only 2.25% (at the lower limit of the interval) is below the limit of the measurement error - such a result can be evaluated as "without effect on strength" and certainly cannot be taken into account in any calculation on real structure. It should therefore be noted that in certain cases (2.25%) the effect on compressive strength of cement mortar is zero in practice.
After incorporating the above 2 comments, I recommend the manuscript for publication.
Round 2
Reviewer 2 Report
All comments and suggestions were appropriately addressed.